# Experimental Investigation on the Behavior of Gravelly Sand Reinforced with Geogrid under Cyclic Loading

**Jia-Quan Wang** [1], **Zhen-Chao Chang** [2], **Jian-Feng Xue** [3,*], **Zhi-Nan Lin** [4,*] **and Yi Tang** [4]

1    College of Civil and Architectural Engineering, Guangxi University of Science and Technology, Guangxi Zhuang Autonomous Region Engineering Research Center of Geotechnical Disaster and Ecological Control, 268 Donghuan Road, Liuzhou 545006, China; wjquan1999@gxust.edu.cn or wjquan1999@163.com
2    Guangxi Beitou Transportation Maintenance Technology Group Co., Ltd., Nanning 530029, China; yhjtchangzc@bgigc.com
3    School of Engineering and IT, University of New South Wales, Campbell, ACT 2612, Australia
4    College of Civil and Architectural Engineering, Guangxi University of Science and Technology, 268 Donghuan Road, Liuzhou 545006, China; tyi-moon@gxust.edu.cn
*    Correspondence: jianfeng.xue@adfa.edu.au (J.-F.X.); zhinan_lin@gxust.edu.cn (Z.-N.L.)

**Abstract:** In view of the dynamic response of geogrid-reinforced gravel under high-speed train load, this paper explores the dynamic characteristics of geogrid-reinforced gravel under semi-sine wave cyclic loading. A number of large scale cyclic triaxial tests were performed on saturated gravelly soil reinforced with geogrid to study the influence of the number of reinforcement layers and loading frequencies on the dynamic responses of reinforced gravelly sand subgrade for high speed rail track. The variation of cumulative axial and volumetric strains, excess pore pressure and resilient modulus with number of loading cycles, loading frequency, and reinforcement arrangement are analyzed. The test results reveal that the cumulative axial strain decreases as the number of reinforcement layers increases, but increases with loading frequency. The resilience modulus increases with the number of reinforcement layers, but decreases as the loading frequency increases. The addition of geogrid can reduce the excess pore water pressure of the sample, but it can slightly enhance the rubber mold embedding effect of the sand sample. As the loading frequency increases, the rubber mold embedding effect gradually weakens.

**Keywords:** geosynthetics; reinforced gravely sand; cyclic loading; large triaxial; high speed train

## 1. Introduction

In recent years, with the vigorous development of high-speed train in China, the amplitude, frequency, and number of loadings on the subgrade have greatly increased. This increases the rate of stiffness degradation and accumulation of deformation of the subgrade filling layer, which in turn increases the stress in the underlying ground layer, resulting in excessive settlement or differential settlement [1].

Much research has been conducted on the behavior of sand under cyclic loading using cyclic triaxial tests, with the focus being on the effects of confining pressure, loading amplitude and frequency, over-consolidation ratio, fine particle content, and relative density, etc. [2–20]. Sharma and Maheshwari [21] conducted cyclic triaxial tests on sands with different degrees of saturation and found that the damping ratio of saturated sand is greater than that of dry sand and partially saturated sand. Jin et al. [22] studied the relative density on the dynamic pore pressure of tailing sand. It is believed that, for dense tailing sand, the growth rate of dynamic pore pressure decreases with the loading frequency and amplitude. Zhang et al. [23] studied the effects of confining pressure, initial shear stress, cyclic stress ratio, and loading frequency on the dynamic strength characteristics of saturated sand in Wenchuan. The authors found that the dynamic strength of saturated sand increases with the increase in confining pressure and loading frequency.

Researchers [24–30] have also studied the dynamic characteristics of materials as pavement subgrade. Leng et al. [31] analyzed the effects of water content, confining pressure and dynamic stress on the dynamic modulus and axial cumulative strain of subgrade coarse-grained materials, and proposed an empirical formula for calculating the dynamic modulus and axial cumulative strain. Sun et al. [32] and Indraratna et al. [33] studied the effect of frequency on the permanent deformation of coarse-grained fillers in railway subgrade. The authors found that the axial strain development mode is plastic stable type at lower loading frequency, and plastic failure type at higher loading frequency. The degree of permanent deformation and deterioration of the roadbed filler increases as the frequency increases.

Research has also been conducted on static and dynamic characteristics of reinforced soils [34–46]. Latha and AM [47] conducted large dynamic triaxial tests on unreinforced and geotextile reinforced sand. It was found that under cyclic loading, with the inclusion of reinforcement, dynamic modulus did not change much at lower confining pressures, but increased significantly at higher confining pressures. Naeini and Gholampoor [48] performed cyclic triaxial tests to study the effect of silt content in geotextile reinforced samples. The results showed that the inclusion of geotextile could decrease the cyclic ductility of dry sand. The sample strength decreased as silt content increase, then decreases once silt content is more than 35%. Qiu et al. [49] performed cyclic triaxial tests on saturated sand, and sand reinforced with polymethyl methacrylate horizontal (H) inclusions and horizontal–vertical inclusions. It is found that both unreinforced and reinforced sand showed stiffness softening characteristics under cyclic loading. The inclusion of reinforcement could improve the dynamic modulus of sand.

In summary, the existing work on reinforced gravel soils is still limited to the following aspects: (1) The research objects are mainly focused on the dynamic characteristics of unreinforced soils, and the use of large triaxial elements to recycle the reinforced soil. Research on behavior is limited, and there are fewer studies on geogrid-reinforced gravel under high-speed train load. (2) The selection of dynamic parameters used in the test of reinforced soil is not suitable for reality, and it is not based on the actual cyclic load in the project. Parameter selection. (3) Window screens, plexiglass, non-woven fabrics, and other alternatives are often used as reinforcement materials, and it is rare to use geogrids as reinforcement materials for testing. (4) The loading waveform of cyclic load is mostly sine wave, which is inconsistent with the actual cyclic load. The carrier shape of the traffic load is quite different. Therefore, in view of the dynamic response of geogrid-reinforced gravel under high-speed train load, this paper explores the dynamic characteristics of geogrid-reinforced gravel under semi-sine wave cyclic loading. The loading parameters are determined based on a typical high speed train with the speed of 300 km/h. The specimens are 150 mm in diameter and 300 mm in height, and are reinforced with a 0-, 1-, 2-, or 3-layer geogrid. The loading frequencies used are 0.5, 1.0, 3.0, and 5.0 Hz. The variation of axial strain, volumetric strain, excess pore water pressure and resilient modulus with loading and reinforcement arrangement is studied. The research results have certain reference significance in the study of the dynamic response of reinforced gravel soil subgrade under cyclic loading.

## 2. Experiment

### 2.1. Materials Used in the Tests

Soils used in subgrade for railway track normally contain gravels [50,51], so a river sand from a river bank in Liuzhou, China, is adopted in the tests. The particle size distribution of the sand is shown in Figure 1. The coefficient of uniformity, $C_u$, is 5.0 and the coefficient of curvature, $C_c$, is 1.25. According to the classification system used in the "Specification of soil test" (SL237-1999) [52], the material is a well-graded gravelly sand. The specific gravity of sand is 2.672, the controlled dry density is 1.77 g/cm$^3$, and the maximum and minimum dry densities are 0.713 and 0.476, respectively. Therefore, the controlled pore ratio of the sample is 0.51, and its relative density is 86.0%, indicating that

the sample is in a dense state. Uniaxial HDPE geogrid with the mesh size 20 mm × 20 mm shown in Figure 2 is used as reinforcement. The properties of the geogrid used in the tests are shown in Table 1.

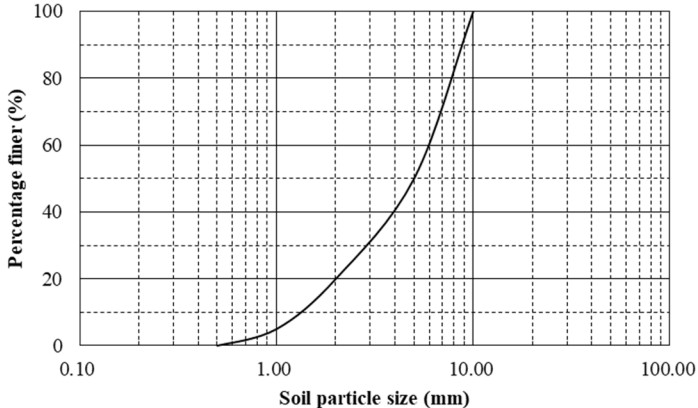

**Figure 1.** Particle size distribution curve.

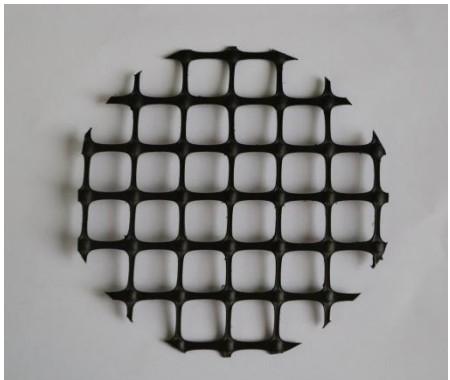

**Figure 2.** Biaxial geogrid.

**Table 1.** Properties of the biaxial geogrid used in the test.

| | |
|---|---|
| The mesh size (mm) | 20.0 × 20.0 |
| Rib width on X-machine direction (mm) | 2.0 |
| Rib width on machine direction (mm) | 3.0 |
| Rib thickness on X-machine direction (mm) | 1.0 |
| Rib thickness on machine direction (mm) | 1.5 |
| Node size (mm) | 4.0 × 3.0 |
| Yield strength (X-machine direction) (kN/m) | 15.4 |
| Yield strength (Machine direction) (kN/m) | 18.6 |

### 2.2. Experiment Setup

A GDS dynamic triaxial test system shown in Figure 3 was used for the tests. The axial loading capacity is 10 kN. The capacity of cell pressure capacity and back pressure is 2 MPa. The loadings can be applied at the frequency up to 5 Hz. The data logging system takes 20 readings for each channel within each loading cycle, for example, 20 axial loadings were taken within each loading cycle.

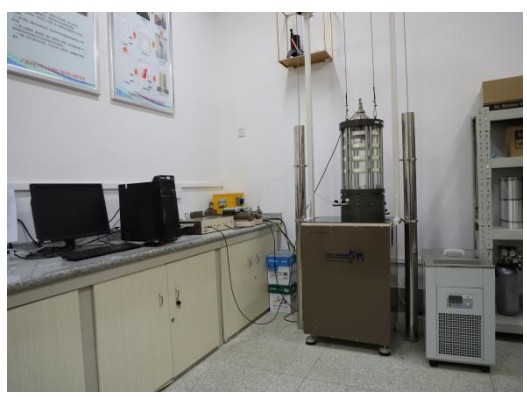

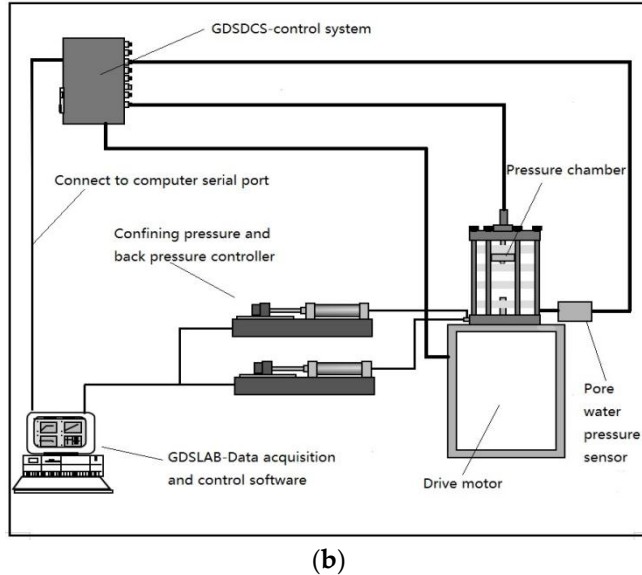

(**a**)                                           (**b**)

**Figure 3.** GDS dynamic triaxial test system: (**a**) physical diagram of the test system; (**b**) schematic diagram of the test system.

### 2.3. Experiment Scheme

To study the influence of the number of reinforcement layers and the loading frequency on the dynamic characteristics of reinforced saturated gravel soil under high-speed train load, consolidated undrained cyclic loading tests were performed on saturated samples with 150 mm diameter and 300 mm high samples as shown in Figure 4. The diameter of the sample was 15 times that of the maximum diameter of the sand sample to minimize the size effect [53]. The reinforcement layers were installed at equal intervals shown in Figure 5. Half sinusoidal loading was used to simulate train traffic [54] as shown in Figure 6. The stopping criteria of the tests are 10,000 loading cycles or 5% of axial strain, whichever is reached first. The GDS dynamic triaxial test system used in this test is controlled by a stress-servo system, which can effectively control the stress amplitude under long-term cyclic loading. During the test, the pre-test is carried out under the same test conditions. The estimated stiffness coefficient in the GDS dynamic triaxial test system is adjusted to stabilize the loading amplitude near the target amplitude, and the loading amplitude will not be large under long-term loading fluctuation. Through the pre-test, the author also verified that loading 10,000 times during the test can ensure that the loading amplitude is stable near the target amplitude without major fluctuations.

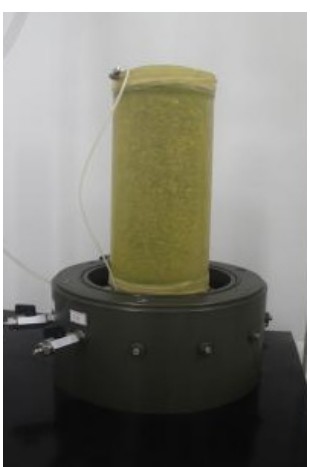

**Figure 4.** A test sample.

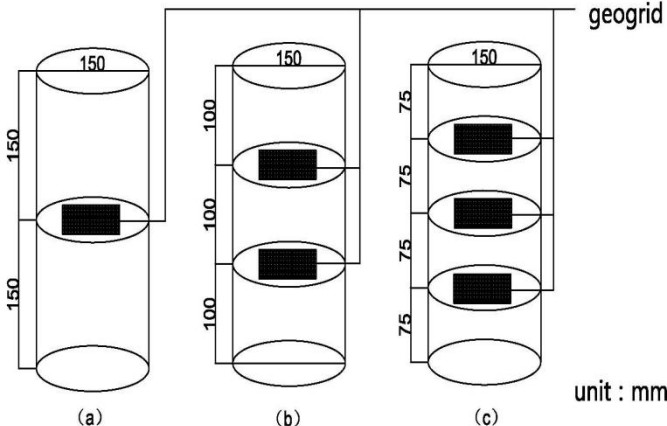

**Figure 5.** Schematic diagram of geogrid layout. (**a**) 1 layer reinforced sample; (**b**) 2 layer reinforced sample; (**c**) 3 layer reinforced sample.

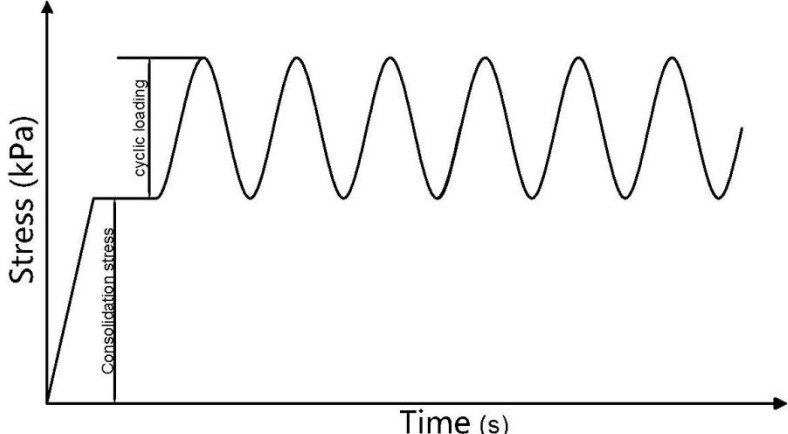

**Figure 6.** Loading waveform.

In the tests, the confining pressure was set as 30 kPa. The value was selected based on the self-weight of soil and rail track. The amplitude of the cyclic loading was determined using the following equation recommended by Zhang [55] for train loading:

$$\sigma_d = 0.26 \times P \times (1 + \alpha v) \tag{1}$$

where $\sigma_d$ is the dynamic stress amplitude at the top surface of the subgrade; P is the static axle load of the train, in the unit of kN; and $(1 + \alpha v)$ is the impact coefficient. For train speed $v = 300{\sim}350$ km/h, $\alpha = 0.003$, and for $v = 200{\sim}250$ km/h, $\alpha = 0.004$. In the tests, the following parameters were adopted: p = 200 kN, $v = 300$ km/h; and $\alpha = 0.003$. Based on this, an amplitude of 100 kPa was adopted for the cyclic loading in the tests.

To consider the effect of reinforcement layers and loading frequency on the response of reinforced soils, the tests shown in Table 2 were performed. In total, 7 sets of samples were tested. The samples were named using loading frequency and reinforcement layers, for example, F1-L1 is the sample with 1 layer (L1) of reinforcement tested under 1Hz loading frequency (F1). All the samples were saturated to a Skempton's B $\geq 0.96$, and consolidated under the isotropic pressure of 30 kPa before applying cyclic loading.

**Table 2.** Test loading parameters and working conditions.

| Test Series | Frequency (Hz) | Number of Reinforcement Layers |
|---|---|---|
| F0.5-L3 | 0.5 | 3 |
| F1-L0 | 1.0 | 0 |
| F1-L1 | 1.0 | 1 |
| F1-L2 | 1.0 | 2 |
| F1-L3 | 1.0 | 3 |
| F3-L3 | 3.0 | 3 |
| F5-L3 | 5.0 | 3 |

## 3. Experiment Results and Analysis of Dynamic Characteristics

*3.1. Analysis of Influence of Reinforced Layer on Dynamic Characteristics of Gravel Soil*

3.1.1. Effect of Number of Reinforced Layers on Axial Cumulative Strain

Figure 7 compares the relationship between the axial cumulative strain εd (the peak strain at the end of each cycle) and loading cycle of the samples with different layers of reinforcement loaded under the same frequency (1 Hz). The comparison shows that the deformation of the samples can be divided into three stages: the fast grow stage during the first 100 cycles, the transition stage between 100 and 1000 (or 2000) cycles, and the stabilization stage after 1000 cycles. A similar pattern was observed by Zhao [1] on the dynamic characteristics of coarse-grained fillers in roadbeds. During the fast-growing stage, the deformation could double after 100 cycles comparing to that after the first loading cycle, and the rate of increment is the highest in the unreinforced sample and the lowest in the sample with three layers of reinforcement. During the transition stage (100 to 1000–2000 cycles), the deformation increment rate decreases, then stabilizes after that. During the stabilization stage, the deformation almost increases with the number of loading cycles at logarithm scale. The unreinforced sample still has the highest increment rate and the other three samples have almost the same rate of increment with loading cycles.

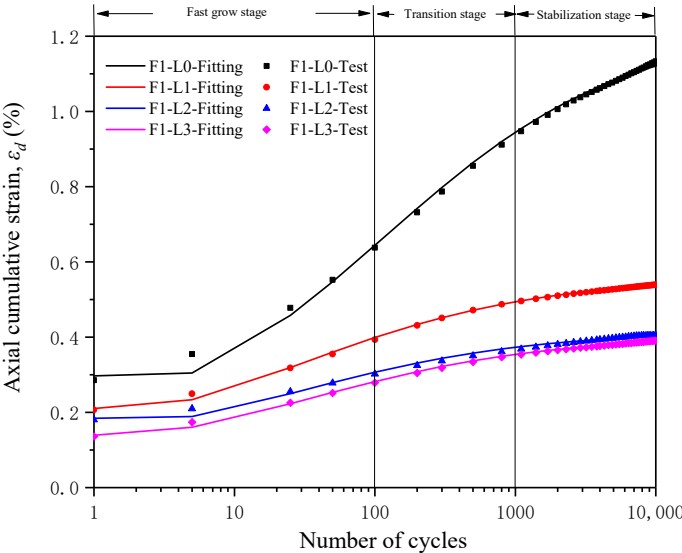

**Figure 7.** Variation of cumulative axial strain with number of loading cycles under different stiffened layers.

Equation (2) is the fitting equation of the axial cumulative strain and the number of loading cycles based on the fitting analysis of the axial cumulative strain and the number of loading cycles under different number of stiffened layers.

$$\varepsilon_d = \frac{N}{a + bN + cN^{0.5}} \qquad (2)$$

where *a*, *b*, and *c* are fitting parameters. The fitted curves using the parameters shown in Table 3 were compared with the test results shown in Figure 7. The comparison suggests that the increment of cumulative axial strain with loading cycles can be fitted with the above equation, regardless of the inclusion of reinforcement. Through comparison, it was found that the values of parameters *a*, *b*, and *c* vary with the number of stiffened layers. When the number of stiffened layers increases, the value of parameter **a** decreases, while the value of parameters *b* and *c* increase. Due to the limited test conditions, only experiments with the number of stiffened layers of 0~3 were conducted, but it was possible to clarify that there is a corresponding relationship between the number of stiffened layers and the parameters *a*, *b*, and *c*; that is, the parameters *a*, *b*, and *c* can be used to characterize the influence of the number of stiffened layers on the axial cumulative strain.

**Table 3.** Test loading parameters and working conditions.

| Number of Reinforcement Layers | Fitting Parameters | | | $R^2$ |
|:---:|:---:|:---:|:---:|:---:|
| | a | b | c | |
| 0 | −5.379 | 0.814 | 7.930 | 0.998 |
| 1 | −4.713 | 1.786 | 7.685 | 0.999 |
| 2 | −6.266 | 2.394 | 9.291 | 0.996 |
| 3 | −6.748 | 2.469 | 11.460 | 0.999 |

3.1.2. Effect of the Number of Reinforced Layers on Resilient Modulus

The variation of resilient modulus (Ed) with number of cycles of the samples is compared in Figure 8. The figure shows that in the first 100 loading cycles, the resilient moduli of the samples remain almost constant, with very slight variation. Overall, the samples with reinforcement are higher in resilient modulus comparing to the non-reinforced sample. The resilient modulus then increases with the number of loading cycles at nearly logarithm scale. For the unreinforced sample, the resilient modulus tends to stabilize after about 8000 cycles, and the resilient modulus increment slows down after that. For the sample with one layer of reinforcement, the resilient modulus increment rate decreases after 8000 cycles. For the other two samples, resilient modulus still increases at nearly logarithm scale after 10,000 cycles. At the end of the tests, the more layers of reinforcement, the higher the resilient modulus. The percentage of increment is higher in the unreinforced sample and the sample with three layers of reinforcement. If we disregard the variation of the resilient modulus of the samples during the first 100 cycles, and normalize resilient moduli of the samples with the values at the 100th loading cycle, (Ed100), as shown in Figure 8b, we can see that the resilient moduli of all the samples increased about 20% at the end of the tests compared to the values at the end of the 100th loading cycle. This may not be used as a conclusion as at the end of the tests, the resilient moduli of the samples still increase with the loading cycle at different rates as shown in the figure.

Based on the analysis of Figures 7 and 8, it can be seen that as the number of loading cycles increases, the overall growth trend of the axial cumulative strain is consistent with the modulus of resilience, and can be divided into a rapid growth phase, a transition phase, and a stable phase. This is due to the gradual increase in the axial cumulative strain of the sample under cyclic loading, and relative displacement between soil particles, and then the soil becomes denser, which results in the soil elastic modulus gradually increasing. With the continuous increase in the cyclic load, the compaction state of the soil tends to be stable, and the changes in the axial cumulative strain and the modulus of resilience also decrease.

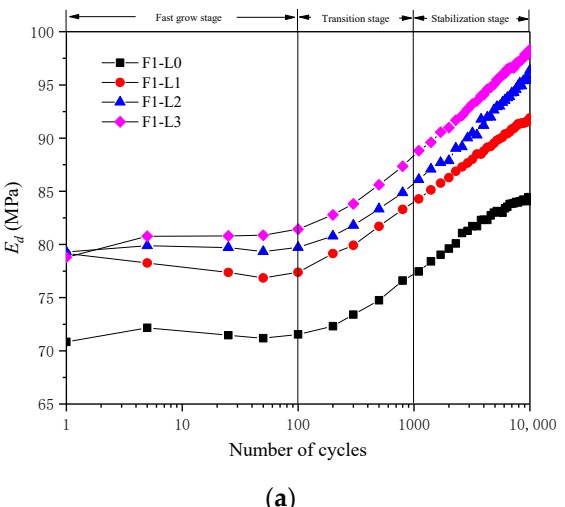

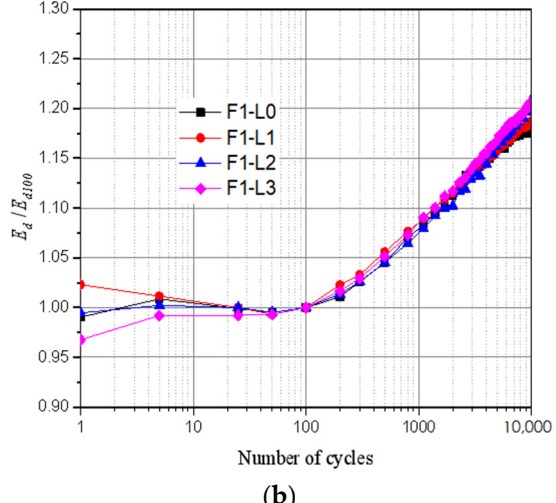

(**a**)                    (**b**)

**Figure 8.** Variation of resilience modulus with number of loading cycles.

### 3.1.3. Effect of Number of Reinforcement Layers on Excess Pore Water Pressure

Figure 9 compares the variation of excess pore water pressure $u_d$ (the peak excess pore water pressure of each cycle) with loading cycles. The figure shows that, unlike accumulative strain, there is no significant transition stage for excess pore water pressure. Excess pore water pressure in the samples increase almost at logarithm scale with the loading cycles, then stabilizes. The number of cycles reaching the stabilization stage differs, but not very much among the samples, for example, about 800 cycles for the unreinforced sample, and 300 cycles for the sample with 2 layers of reinforcement. After 1000 cycles, excess pore water pressure in all samples stabilized, and the value is the highest in the unreinforced sample, which is about 16 kPa, and the lowest in the sample with two layers of reinforcement at a value of about 11 kPa. This indicates that the inclusion of geogrid may be beneficial in terms of excess pore water pressure dissipation.

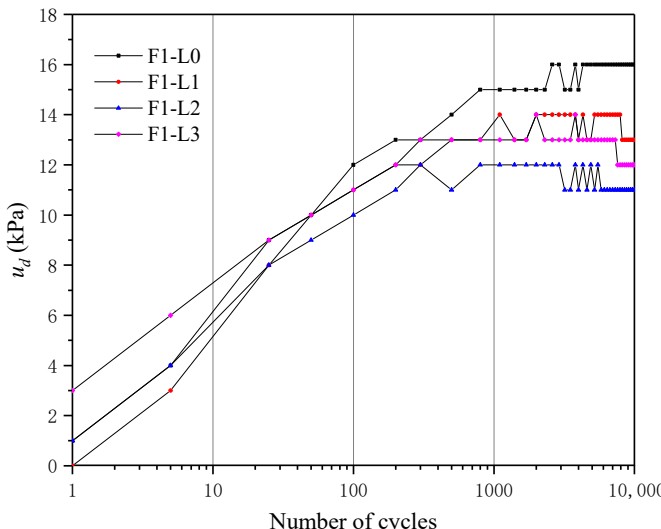

**Figure 9.** Relationship between dynamic pore water pressure and number of loading cycles.

### 3.1.4. Effect of Number of Reinforced Layers on Volumetric Strain

During the test, when the volumetric strain is positive, it means the sample volume expands, while the sample volume shrinks when it is negative. During the undrained test, the increase in volume strain is caused by the embedding effect of the sand sample membrane. Some scholars have found that when the sand sample is consolidated, the

rubber membrane will be embedded in the pores of the surface of the sample, and the greater the effective stress, the greater the amount of rubber membrane is embedded. During undrained shearing, as the pore water pressure increases, the effective stress decreases, and the embedded amount of the rubber membrane decreases; that is, part of the water in the sample squeezes out the rubber membrane, resulting in the sample under undrained conditions. The phenomenon of volumetric strain growth occurs [56].

Figure 10 describes the variation of volumetric strain $\varepsilon_v$ with the number of loading cycles. The comparison shows that the inclusion of geogrid could slightly increase the volumetric strain of the samples but would not change the overall relationship: where the volumetric strain of the samples increases with loading cycles. It shows that the addition of geogrid can slightly aggravate the embedding effect of sand samples, and it will gradually appear as the number of cyclic load loading increases.

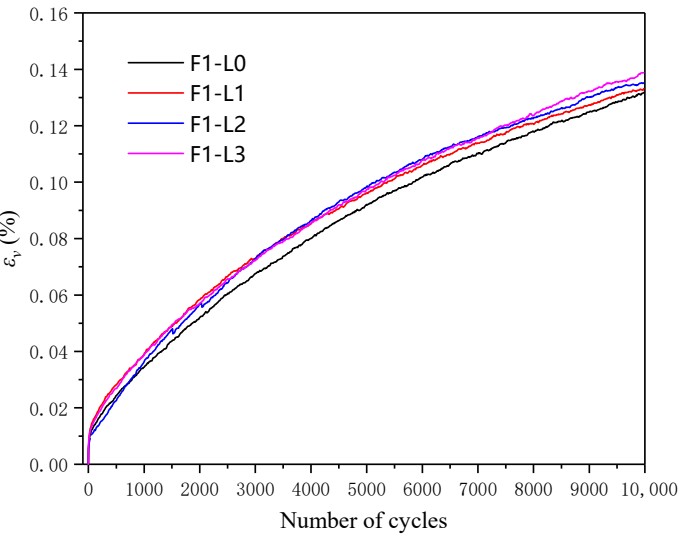

**Figure 10.** Relationship between volumetric strain and number of loading cycles.

3.1.5. Reinforcing Effect Coefficients of Different Reinforcing Schemes

The following reinforcing effect coefficients are used to compare the efficiency of different reinforcing schemes:

$$\eta_{\varepsilon d} = \frac{(\varepsilon_r - \varepsilon_0)}{\varepsilon_0} \tag{3}$$

$$\eta_{Ed} = \frac{(E_r - E_0)}{E_0} \tag{4}$$

$$\eta_{ud} = \frac{(u_r - u_0)}{u_0} \tag{5}$$

$$\eta_{vd} = \frac{(\varepsilon_{vr} - \varepsilon_{v0})}{\varepsilon_{v0}} \tag{6}$$

where $\eta_{\varepsilon d}$, $\eta_{Ed}$, $\eta_{ud}$, $\eta_{vd}$ are reinforcing effect coefficients in respect of axial cumulative strain, resilience modulus, excess pore water pressure, and volumetric accumulative strain respectively; $\varepsilon_0$, $E_0$, $u_0$, and $\varepsilon_{v0}$ are the axial cumulative strain, resilient modulus, excess pore water pressure, and volumetric accumulative strain of the unreinforced sample; and $\varepsilon_r$, $E_r$, $u_r$, and $\varepsilon_{vr}$ are the axial cumulative strain, resilient modulus, excess pore water pressure, and volumetric accumulative strain of the reinforced sample. Since all the tests finished after 10,000 cycles, the values are calculated using the test results of the last loading cycle.

The reinforcing effect coefficients of the three reinforced samples loaded under 1 Hz are compared in Table 4. The results show that, with the inclusion of reinforcement,

the accumulated axial strain and the accumulated excess pore water pressure decrease significantly. When including two layers of reinforcement, the accumulated axial strain reduced about 64.3%, which is slightly lower than that of the sample with three layers of reinforcement (65.5%), but the reduction in excess pore water pressure is about 31.3%, which is the highest of the three samples. The reinforcing effect coefficient in respect of resilient modulus increased from 8.8% to 14.1% and 16.4% by including 1, 2 and 3 layers of reinforcement, comparing to the sample with no reinforcement. Considering the combined effect, the inclusion of two layers of reinforcement is the most optimum in terms of cost and improvement of performance. The addition of geogrid can reduce the excess pore water pressure of the sample and increase slightly the embedding effect of the sand sample. Furthermore, the embedding effect gradually weakens as the loading frequency increases.

**Table 4.** Reinforcement effect coefficient under different numbers of reinforcement layers.

| Number of Reinforcement Layers | $\eta_{\varepsilon d}$ | $\eta_{Ed}$ | $\eta_{ud}$ | $\eta_{vd}$ |
| --- | --- | --- | --- | --- |
| 0 | 0.000 | 0.000 | 0.000 | 0.000 |
| 1 | −0.524 | 0.088 | −0.188 | 0.010 |
| 2 | −0.643 | 0.141 | −0.313 | 0.026 |
| 3 | −0.655 | 0.164 | −0.250 | 0.053 |

### 3.2. Influence of Loading Frequency on Reinforced Gravelly Soil

To study the effect of loading frequency on the behavior of reinforced samples, four samples with three layers of reinforcement were loaded at 0.5 Hz, 1 Hz, 3 Hz, and 5 Hz. The samples were prepared using the same procedure to the designed relative density. The results are discussed in the following.

#### 3.2.1. Effect of Loading Frequency on Cumulative Axial Strain

Figure 11 shows the $\varepsilon_d$–$N$ curve of the samples under different loading frequencies. Overall the cumulative axial strain increases with number of loading cycles. At the end of the 100th cycle, the axial cumulative strain of the specimen increases with the increase in the number of cycles, and the axial cumulative strain is not significantly affected by the frequency. When the cyclic loading is between 100 and 1000 times, the axial cumulative strain of the specimen increases at different rates, and the higher the frequency, the greater the axial cumulative strain. When the cycle numbers larger than 1000 cycles, the axial cumulative strain of the sample increases almost logarithmically. The growth rate of the axial cumulative strain under the loading frequency of 3 Hz and 5 Hz is similar and both higher than that under the loading frequency of 0.5 Hz and 1 Hz. This stage continues to show the law that the higher the frequency, the greater the axial cumulative strain. Based on the observation, we can see that the higher the loading frequency the greater the cumulative axial strain. This suggests that train speed and time intervals between the trains are important factors to be considered in the long-term settlement prediction of the subgrade of railway. The curves of the cumulative axial strain can be fitted using the following Equation:

$$\varepsilon_d = \frac{a}{1 + b \times N^c} \tag{7}$$

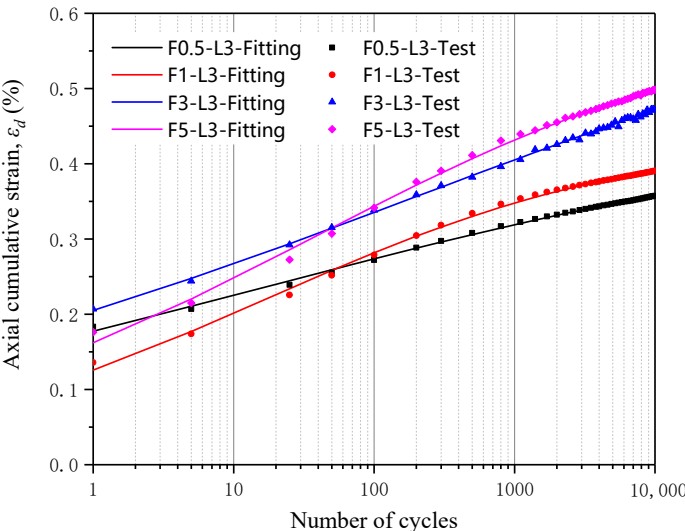

**Figure 11.** Variation of cumulative axial strain with number of loading cycles under different loading frequencies.

The fitting parameters of the curves are show in Table 5.

**Table 5.** Cumulative axial strain fitting parameters at loading different frequencies.

| Frequency (Hz) | Fitting Parameters | | | $R^2$ |
|---|---|---|---|---|
| | a | b | c | |
| 0.5 | −6.326 | 2.764 | 8.887 | 0.994 |
| 1.0 | −6.747 | 2.469 | 11.46 | 0.999 |
| 3.0 | −6.397 | 2.091 | 8.972 | 0.983 |
| 5.0 | −6.887 | 1.933 | 10.470 | 0.997 |

### 3.2.2. Effect of Loading Frequency on Resilient Modulus

The $E_d$–$N$ curve of the samples loaded under different frequencies is shown in Figure 12. Similar to the results shown in Figure 8, the resilient moduli of the samples vary slightly during the first 100 loading cycles. Between 100 and 1000 loading cycles, the resilient moduli of the samples increase rapidly, then slow down after that. After 1000 loading cycles, the resilient moduli increase nearly logarithmically with the loading cycles. Figure 8a shows that the lower the loading frequency the higher the resilient modulus of a sample. This is opposite to that observed by Cai et al. (2017) on cyclic behavior of coarse-grained subgrade. This may be due to the material difference (particle size distribution) and the inclusion of reinforcement. This suggests that train speed and interval between trains may affect the resilient modulus of the subgrade of tracks. By normalizing the resilient modulus of each sample with its maximum value, we can see that after about 4000 loading cycles, the resilient moduli of the samples increase with loading cycles almost at the same rate as shown in Figure 8b. This suggests that loading frequency may not affect the resilient increment rate in long term.

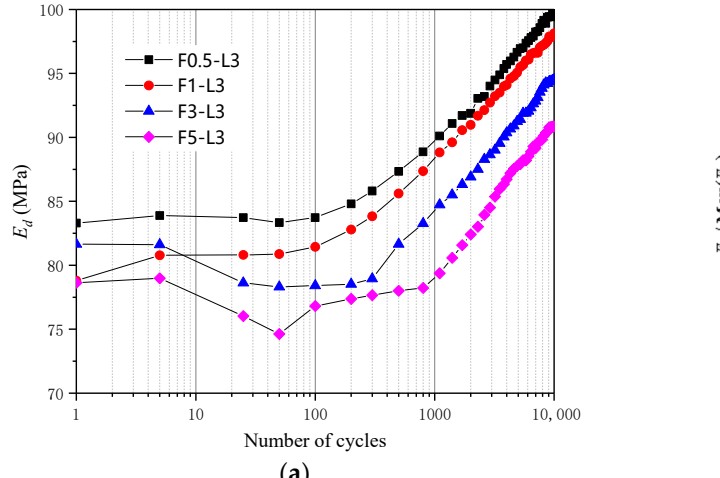
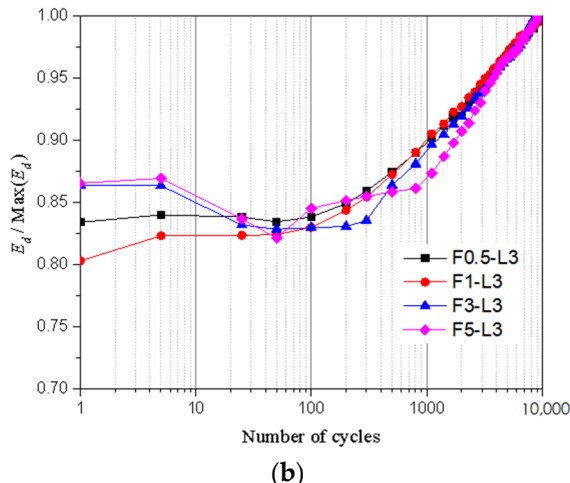

**Figure 12.** Variation of (**a**): resilient modulus and (**b**): normalized resilient modulus with number of loading cycles.

### 3.2.3. Effect of Frequency on Excess Pore Water Pressure

Figure 13 shows the $u_d$–$N$ curves of the samples loaded under different frequencies. The comparison shows that excess pore water pressure in the samples increase nearly logarithmically with loading cycles before stabilizing. The higher the loading frequency, the longer it takes the excess pore water pressure to stabilize, and the higher the excess pore water pressure. As loading frequency increases from 0.5 Hz to 5 Hz, the excess pore water pressure ratio of the samples increases from 0.3 to 0.57.

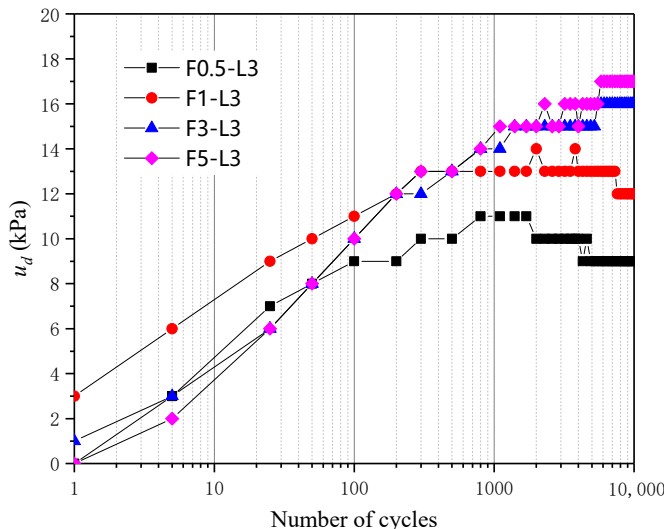

**Figure 13.** Variation of excess pore water pressure with number of loading cycles.

### 3.2.4. Effect of Frequency on Accumulative Volumetric Strain

Figure 14 compares the $\varepsilon_v$–$N$ curves of the samples loaded under different frequencies. It shows that loading frequency has great effect on accumulative volumetric strain of the samples: the higher the frequency, the lower the accumulative volumetric strain. Under higher loading frequencies, the volumetric strain increases at lower rate with loading cycles. This shows that the lower the frequency, the more significant the rubber membrane embedding effect of the sand sample is. When the frequency is 1 Hz and 5 Hz, the volumetric strain of the sample under 10,000 cyclic loads is 0.156% and 0.057%, which is a difference of 0.099%.

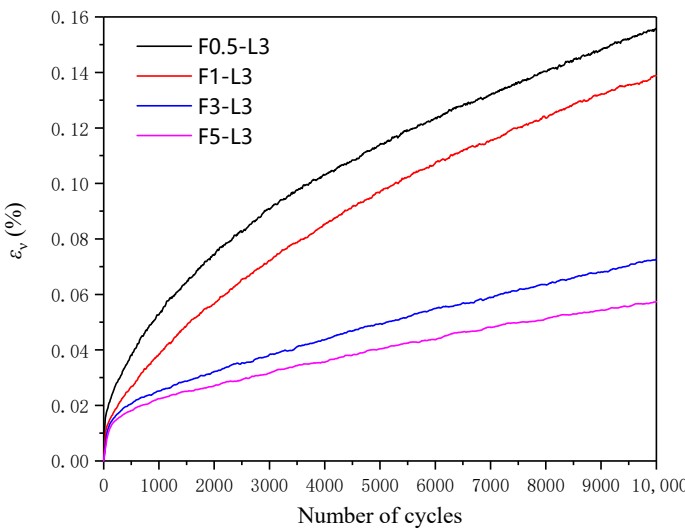

**Figure 14.** Variation of volumetric strain with number of loading cycles.

*3.3. Discussion*

Under the same loading frequency, there is not much of a reduction in cumulative strain comparing the sample with three layers of geogrid to that with two layers of geogrid. This suggests that there is an optimum number of reinforcement layers in terms of cost and efficiency. Under the action of different loading frequencies, the regularity of the axial cumulative strain of the specimens affected by frequency before 100 cycles is not obvious, but with the increase of the number of cyclic load loading, the cumulative axial strain of the specimen will show the law that the higher the loading frequency, the greater the cumulative axial strain. It shows that under long-term cyclic load, the frequency of train load is an important factor that needs to be considered in the long-term settlement prediction of railway subgrade. After large number of loading cycles, 4000 cycles in the tests, the resilient moduli of the samples with the same number of reinforcement layers increase nearly at the same rate with the loading cycles. This may suggest that in long-term, the rate of increment of resilient modulus is not affect by loading frequency.

**4. Conclusions**

A set of cyclic loading tests were performed on seven gravelly sand samples with the diameter of 150 mm and 300 mm high. The samples were reinforced with 0, 1, 2 and 3 layers of geogrid. The samples were loaded under cyclic loading at different frequencies to study the effect of number of reinforcement layers and loading frequency on cyclic behavior of the samples. The main conclusions are as follows:

(1) The inclusion of geogrid could reduce the cumulative axial strain of the samples under cyclic loading. The cumulative axial strain increases with loading frequency. The cumulative axial strain of the samples tends to stabilize after about 2000 loading cycles. After that the cumulative axial strain increases with loading cycles at nearly logarithm scale.

(2) The resilient moduli of the samples increase with loading cycles. For the sample with fewer reinforcement layers, the increment rate decreases faster than those with more layers of reinforcement. With the same number of reinforcement layers, the higher the loading frequency, the lower the resilient modulus.

(3) The inclusion of geogrid may reduce the excess pore water pressure developed in the samples: the more layers of reinforcement, the less the excess pore water pressure. For reinforced samples, the higher the loading frequency, the greater the excess pore water pressure.

(4) Under the same loading frequency, the addition of geogrid can slightly aggravate the rubber mold embedding effect of sand samples. For samples with the same number of

reinforcement layers, the lower the loading frequency, the more significant the rubber mold embedding effect of the sand samples.

**Author Contributions:** Paper revision and polishing, J.-Q.W. and J.-F.X.; Laboratory experiment, Z.-C.C. and Z.-N.L.; Paper writing, Z.-C.C.; Literature research, Y.T. All authors have read and agreed to the published version of the manuscript.

**Funding:** The project was funded by the National Natural Science Foundation of China (No. 41962017), the Natural Science Foundation in Guangxi Province of China (No. 2017GXNSFAA198170, No. 2021GXNSFBA196043), the High Level Innovation Team and Outstanding Scholars Program of Guangxi Institutions of Higher Learning of China (GuiJiaoRenCai [2020]6), the doctoral Foundation of Guangxi University of Science and Technology (No. 03200009), the Project to Enhance the Basic Research Ability of Young and Middle-aged Teachers in Guangxi Universities (No. 2020KY08023) and the ARC Industry Transformation Research Hub (IH180100010-IH18.09.1).

**Data Availability Statement:** The data presented in this study are available on request from the corresponding author.

**Conflicts of Interest:** The authors declare no conflict of interest.

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
