# Peer review of "Experimental Investigation on the Behavior of Gravelly Sand Reinforced with Geogrid under Cyclic Loading"

_applsci, doi:10.3390/app112412152_

Round 1
Reviewer 1 Report
The submitted paper is concerned with the analysis of the behaviour of sand reinforced with geogrids under cyclic loading by means of a laboratory campaign of cyclic undrained triaxial tests.
The triaxial tests are performed on samples of gravelly sand with a D50 of 5 m with up to 3 layers of geogrid. The choice of the compaction level with a Dr of 60% is not justified and is probably not adapted to the question analysed by the paper (behaviour of railroad tracks).
The tests are performed in undrained conditions with a confining stress of 30kPa and a maximum cyclic vertical load of 100kPa. The confining stress is low compared to the height of the sample (0.3 m) which induces a gradient of stress that is not negligible and that could affect the tests results.
The loading is cyclic with a maximum of 10000 cycles with a frequency ranging from 0.5 to 5Hz. The question of accurate control of the applied load during this type of loading is not analysed by the authors.
The authors do not justify the choice of undrained conditions to determine the effect of cyclic loads supposed to be applied by the passage of high speed trains: if during the passage of the train, it can be relevant to assume that the soil behaves in undrained conditions, it is clearly not the case between 2 train passages, period during which the excess pore pressure can easily dissipate (the soil being highly permeable).
The main problem of the paper relates to the obtained results: during the cycles, it appears that there is a development of pore pressure and in the mean time there is a volumetric strain that also increases (by the way, the sign convention should be mentioned: is it contractancy or dilatancy that arises).
The reviewer therefore believes that there are technical problems during the tests. The obtained results are thus questionable.
It is proposed to reject the paper for that reason unless the authors are able to justify these results.
Author Response
1)The triaxial tests are performed on samples of gravelly sand with a D50 of 5 m with up to 3 layers of geogrid. The choice of the compaction level with a Dr of 60% is not justified and is probably not adapted to the question analysed by the paper (behaviour of railroad tracks).
Reply: The authors sincerely thank the reviewer for this constructive suggestion for improving the manuscript. After repeated confirmation, the authors found that there was an error in the control pore ratio calculation at this place, which has been revised and supplemented in lines 99-102 on page 2 and 3. The supplementary content is as follows: “The specific gravity of sand is 2.672, the controlled dry density is 1.77g/cm3, and the maximum and minimum dry densities are 0.713 and 0.476, respectively. Therefore, the controlled pore ratio of the sample is 0.51, and its relative density is 86.0%, indicating that the sample is in a dense state.”
2)The tests are performed in undrained conditions with a confining stress of 30kPa and a maximum cyclic vertical load of 100kPa. The confining stress is low compared to the height of the sample (0.3 m) which induces a gradient of stress that is not negligible and that could affect the tests results.
Reply: The authors sincerely thank the reviewer for this constructive suggestion for improving the manuscript. The confining pressure in this test is calculated based on the self-weight stress of the railway subgrade in the actual project, so the confining pressure strength is low. In the following research, we will conduct a series of test on the behavior of gravelly sand reinforced with geogrid under high confining and cyclic loading.
3)The loading is cyclic with a maximum of 10000 cycles with a frequency ranging from 0.5 to 5Hz. The question of accurate control of the applied load during this type of loading is not analysed by the authors.
Reply: The authors appreciate the reviewer’s recommendation. According to this comment, the following sentences have been added in Page 4, Line 126-133 to explain this:
“The GDS dynamic triaxial test system used in this test is controlled by a stress-servo system, which can effectively control the stress amplitude under long-term cyclic loading. During the test, the pre-test is carried out under the same test conditions. The estimated stiffness coefficient in the GDS dynamic triaxial test system is adjusted to stabilize the loading amplitude near the target amplitude, and the loading amplitude will not be large under long-term loading fluctuation. Through the pre-test, the author also verified that loading 10,000 times during the test can ensure that the loading amplitude is stable near the target amplitude without major fluctuations.”
4)The authors do not justify the choice of undrained conditions to determine the effect of cyclic loads supposed to be applied by the passage of high speed trains: if during the passage of the train, it can be relevant to assume that the soil behaves in undrained conditions, it is clearly not the case between 2 train passages, period during which the excess pore pressure can easily dissipate (the soil being highly permeable).
Reply: The authors sincerely thank the reviewer for this constructive suggestion for improving the manuscript. The dynamic load used in this study is generated when two adjacent train wheels pass through the same position on the track. Since the train speed is as high as 300km/h and the distance between the two rows of wheels is relatively short, the frequency of cyclic loading is very high. As a result, the pore water pressure was too late to dissipate, so the undrained condition was selected for the test.
5)The main problem of the paper relates to the obtained results: during the cycles, it appears that there is a development of pore pressure and in the mean time there is a volumetric strain that also increases (by the way, the sign convention should be mentioned: is it contractancy or dilatancy that arises).
Reply: The authors sincerely thank the reviewer for this constructive suggestion for improving the manuscript. The author has repeatedly checked the test procedures, test principles, and test conclusions in this study. For the volume strain part, the expression may not be clear enough. The positive value of the volumetric strain expressed in this paper is the strain of volume reduction, that is, the larger the volumetric strain, the more volumetric shrinkage. Therefore, the author supplemented the sign convention of volume strain on page 9, line 251-253, when the value is positive means volume contraction, while negative value means volume expansion. The volumetric strain increases, the sample volume decreases, and the pore water pressure increases, which conforms to the general law.

Reviewer 2 Report
- Figure 7 : Add the line to divide the 3 zones (the fast grow stage, the transition stage and stabilization stage) in the Figure 7.
- Figure 8 : Add the line to divide the 3 zones (the fast grow stage, the transition stage and stabilization stage) in the Figure 8.
- Based on 3 zones in Figure 7 and Figure 8, explain the relationship between axial cumulative strain and resilient modulus.
- Figure 7 and Figure 11 have the duplicate name. It should be change to difference name.
- Based on equation 1 in line 123 used for train loading, could the author be sure that it can be applied for high speed train in this study?
Author Response
1)Figure 7 : Add the line to divide the 3 zones (the fast grow stage, the transition stage and stabilization stage) in the Figure 7.
Reply: The authors sincerely thank the reviewer for this constructive suggestion for improving the manuscript. The content of Figure 7 has been revised on page 6, line 173 based on the comment.
2)Figure 8 : Add the line to divide the 3 zones (the fast grow stage, the transition stage and stabilization stage) in the Figure 8.
Reply: The authors sincerely thank the reviewer for this constructive suggestion for improving the manuscript. The content of Figure 8 has been revised on page 8, line 229 based on the comment.
3)Based on 3 zones in Figure 7 and Figure 8, explain the relationship between axial cumulative strain and resilient modulus.
Reply: The authors appreciate the reviewer’s recommendation. According to this comment, the following sentences have been added in Page 8, Line 220-228 to explain this:
“Based on the analysis of Figures 7 and 8, it can be seen that as the number of loading cycles increases, the overall growth trend of the axial cumulative strain is consistent with the modulus of resilience, and can be divided into a rapid growth phase, a transition phase, and a stable phase. This is due to the gradual increase in axial cumulative strain of the sample under cyclic loading, and relative displacement between soil particles, and then the soil becomes denser, which result to the soil elastic modulus gradually increase. With the continuous increase of the cyclic load, the compaction state of the soil tends to be stable, and the changes in the axial cumulative strain and the modulus of resilience also decrease.”
4)Figure 7 and Figure 11 have the duplicate name. It should be change to difference name.
Reply: According to this comment, the title of Figure 7 in Page 7, Line 178-179 was revised to "The variation of the axial cumulative strain with the number of loading cycles under different stiffened layers." The title of Figure 11 in Page 11, Line 324-325 was modified as "The change of the axial cumulative strain with the number of loading cycles under different loading frequencies".
5)Based on equation 1 in line 123 used for train loading, could the author be sure that it can be applied for high speed train in this study?
Reply: The authors sincerely thank the reviewer for this constructive suggestion for improving the manuscript. Formula 1 is to calculate the cyclic load amplitude of the train by selecting different impact coefficients in different speed ranges. In this study, the train speed is 300km/h, the impact coefficient α=0.003 is selected, and the stress amplitude is calculated to be 98.8kPa, so the cyclic load amplitude was set to 100kPa in the test. Therefore, according to the engineering background of the high-speed train in this study, Formula 1 is applicable.

Reviewer 3 Report
Dear Authors,
Thank you for your manuscript, here will be following comments:
- Abstract doesn’t provide sufficient information on the novelty of your study.
- Introduction has to be more elaborated from the point what is done so far and what you suggest to improve with underling the novelty of your study.
- In your results you need to emphasize the comparison of your results and so far available now provided by other researchers.
- Your conclusions are too long, those should be rewritten in a bullet form and with exact definition of the novelty of your study.
Author Response
1)Abstract doesn`t provide sufficient information on the novelty of your study.
Reply: The authors sincerely thank the reviewer for this constructive suggestion for improving the manuscript. This has been revised in page 1 lines 18-19.
2)Introduction has to be more elaborated from the point what is done so far and what you suggest to improve with underling the novelty of your study.
Reply: The authors sincerely thank the reviewer for this constructive suggestion for improving the manuscript. This has been modified in the introduction In page 2 lines 74-93.
“In summary, the existing work on reinforced gravel soils is still limited to the following aspects: (1) The research objects are mainly focused on the dynamic characteristics of unreinforced soils, and the use of large triaxial elements to recycle the reinforced soil Research on behavior is limited, and there are fewer studies on geogrid-reinforced gravel under high-speed train load; (2) The selection of dynamic parameters used in the test of reinforced soil is not suitable for reality, and it is not based on the actual cyclic load in the project. Parameter selection; (3) Window screens, plexiglass, non-woven fabrics and other alternatives are often used as reinforcement materials, and it is rare to use geogrids as reinforcement materials for testing; (4) The loading waveform of cyclic load is mostly sine wave, which is inconsistent with actual The carrier shape of the traffic load is quite different. Therefore, in view of the dynamic response of geogrid-reinforced gravel under high-speed train load, this paper explores the dynamic characteristics of geogrid-reinforced gravel under semi-sine wave cyclic loading. The loading parameters are determined based on a typical high speed train with the speed of 300 km/h. The specimens are 150 mm in diameter and 300 mm in height, and are reinforced with 0, 1, 2 or 3 layers geogrid. The loading frequencies used are 0.5, 1.0, 3.0, and 5.0 Hz. The variation of axial strain, volumetric strain, excess pore water pressure and resilient modulus with loading and reinforcement arrangement is studied. The research results have certain reference significance in the study of the dynamic response of reinforced gravel soil subgrade under cyclic loading.”
3)In your results you need to emphasize the comparison of your results and so far available now provided by other researchers.
Reply: The authors sincerely thank the reviewer for this constructive suggestion for improving the manuscript. This comment has been supplemented on page 6, lines 167-168, page 9, lines 256-259, and page 12, lines 341-343.
4)Your conclusions are too long, those should be rewritten in a bullet form and with exact definition of the novelty of your study.
Reply: The authors sincerely thank the reviewer for this constructive suggestion for improving the manuscript. This has been revised in the conclusions.
3.3. Discussion
A set of cyclic loading tests were performed on 7 gravelly sand samples with the diameter of 150 mm and 300 mm high. The samples were reinforced with 0, 1, 2 and 3 layers of geogrid. The samples were loaded under cyclic loading at different frequencies to study the effect of number of reinforcement layers and loading frequency on cyclic behavior of the samples.
With the limited number of tests performed, it was found that: Under the same loading frequency, there is not much of reduction in cumulative strain comparing the sample with three layers of geogrid to that with two layers of geogrid. This suggests that there is an optimum number of reinforcement layers in terms of cost and efficiency. Under the action of different loading frequencies, the regularity of the axial cumulative strain of the specimens affected by frequency before 100 cycles is not obvious, but as the number of loading cycles increases, the cumulative axial strain increases with increasing frequency the regularity. It shows that under long-term cyclic load, the frequency of train load is an important factor that needs to be considered in the long-term settlement prediction of railway subgrade. After large number of loading cycles, 4000 cycles in the tests, the resilient moduli of the samples with the same number of reinforcement layers increase nearly at the same rate with loading cycles. This may suggest that in long term, the rate of increment of resilient modulus is not affect by loading frequency.
- Conclusions
(1)The inclusion of geogrid could reduce the cumulative axial strain of the samples under cyclic loading. The cumulative axial strain increases with loading frequency. The cumulative axial strain of the samples tends to stabilize after about 2000 loading cycles. After that the cumulative axial strain increases with loading cycles at nearly logarithm scale.
(2)The resilient moduli of the samples increase with loading cycles. For the sample with less number of reinforcement layers, the increment rate decreases faster than those with more layers of reinforcement. With the same number of reinforcement layers, the higher the loading frequency, the lower the resilient modulus.
(3)The inclusion of geogrid may reduce the excess pore water pressure developed in the samples: the more layers of reinforcement, the less the excess pore water pressure. For reinforced samples, the higher the loading frequency, the greater the excess pore water pressure.
(4)Under the same loading frequency, the inclusion of geogrid has little influence on the volumetric strain development in the samples. For the samples with the same number of reinforcement layers, the higher the loading frequency, the lower the volumetric strain and its increment rate with loading cycles.

Reviewer 4 Report
This paper investigates an interesting topic from the previous large scale cyclic triaxial tests performed on saturated gravelly soil reinforced with geogrid to study the influence of the number of reinforcement layers and loading frequencies on the dynamic responses of reinforced gravelly sand subgrade for high speed rail track. In particular, the paper studies the behavior of geogrid reinforced gravely sand subjected to half sine wave cyclic loading.
The methodology is correct and the structure of the paper pertinent. English is also fine. Some observations need to be considered: 1. Equation 2 needs to be explained and the references which proposed needed to be cited. Parameters a, b, c need to be explained as well. 2. Figure 9 is not clear. Please scale the x-axis. 3. Eq. 8 and figure 11 need the same as observation 1 and 2, respectively. Conclusion This section needs to be splitted in two sections: discussion and conclusions. Where conclusions are specifically focused on demonstrating: the originality, the tasks, and the fullfillment of the tasks.
Author Response
1) Equation 2 needs to be explained and the references which proposed needed to be cited. Parameters a, b, c need to be explained as well.
Reply: The authors appreciate the reviewer’s recommendation.
According to this comment, the following sentences have been added in Page 7, Line 176-178. “Equation 2 is the fitting equation of the axial cumulative strain and the number of loading cycles based on the fitting analysis of the axial cumulative strain and the number of loading cycles under different number of stiffened layers.”
And the following sentences have been added in Page 7, Line 191-199.
“Through comparison, it is found that the values of parameters a, b, and c vary with the number of stiffened layers, When the number of stiffened layers increases, the value of parameter a decreases, while the value of parameters b and c increase. Due to the limited test conditions, only experiments with the number of stiffened layers of 0~3 have been conducted, but it has been possible to clarify that there is a corresponding relationship between the number of stiffened layers and the parameters a, b, and c, that is, the parameters a, b and c can be used to characterize the influence of the number of stiffened layers on the axial cumulative strain.”
2) Figure 9 is not clear. Please scale the x-axis.
Reply: The authors appreciate the reviewer’s recommendation. According to this comment, on page 9, line 249, Figure 9 is modified and replaced.
3) Eq.8 and figure 11 need the same as observation 1 and 2, respectively.
Reply: The authors appreciate the reviewer’s recommendation. According to this comment, on page 11, lines 309-317 have been modified, and the specific content is as follows:
"At the end of the 100th cycle, the axial cumulative strain of the specimen increases with the increase in the number of cycles, and the axial cumulative strain is not significantly affected by the frequency. When the cyclic loading is between 100 and 1000 times, the axial cumulative strain of the specimen increases at different rates, and the higher the frequency, the greater the axial cumulative strain. When the cycle numbers larger than 1000 cycles, the axial cumulative strain of the sample increases almost logarithmically. The growth rate of the axial cumulative strain under the loading frequency of 3Hz and 5Hz is similar and both higher than that under the loading frequency of 0.5Hz and 1Hz. This stage continues to show the law that the higher the frequency, the greater the axial cumulative strain.”
4) Conclusion This section needs to be splitted in two sections: discussion and conclusions. Where conclusions are specifically focused on demonstrating: the originality, the tasks, and the fullfillment of the tasks.
Reply: The authors sincerely thank the reviewer for this constructive suggestion for improving the manuscript. The author has reorganized and revised the conclusion part. According to this comment, on page 13, lines 361-393 have been modified, and the specific content is as follows:
3.3. Discussion
A set of cyclic loading tests were performed on 7 gravelly sand samples with the diameter of 150 mm and 300 mm high. The samples were reinforced with 0, 1, 2 and 3 layers of geogrid. The samples were loaded under cyclic loading at different frequencies to study the effect of number of reinforcement layers and loading frequency on cyclic behavior of the samples.
With the limited number of tests performed, it was found that: Under the same loading frequency, there is not much of reduction in cumulative strain comparing the sample with three layers of geogrid to that with two layers of geogrid. This suggests that there is an optimum number of reinforcement layers in terms of cost and efficiency. Under the action of different loading frequencies, the regularity of the axial cumulative strain of the specimens affected by frequency before 100 cycles is not obvious, but as the number of loading cycles increases, the cumulative axial strain increases with increasing frequency the regularity. It shows that under long-term cyclic load, the frequency of train load is an important factor that needs to be considered in the long-term settlement prediction of railway subgrade. After large number of loading cycles, 4000 cycles in the tests, the resilient moduli of the samples with the same number of reinforcement layers increase nearly at the same rate with loading cycles. This may suggest that in long term, the rate of increment of resilient modulus is not affect by loading frequency.
- Conclusions
(1) The inclusion of geogrid could reduce the cumulative axial strain of the samples under cyclic loading. The cumulative axial strain increases with loading frequency. The cumulative axial strain of the samples tends to stabilize after about 2000 loading cycles. After that the cumulative axial strain increases with loading cycles at nearly logarithm scale.
(2) The resilient moduli of the samples increase with loading cycles. For the sample with less number of reinforcement layers, the increment rate decreases faster than those with more layers of reinforcement. With the same number of reinforcement layers, the higher the loading frequency, the lower the resilient modulus.
(3) The inclusion of geogrid may reduce the excess pore water pressure developed in the samples: the more layers of reinforcement, the less the excess pore water pressure. For reinforced samples, the higher the loading frequency, the greater the excess pore water pressure.
(4) Under the same loading frequency, the inclusion of geogrid has little influence on the volumetric strain development in the samples. For the samples with the same number of reinforcement layers, the higher the loading frequency, the lower the volumetric strain and its increment rate with loading cycles.

Round 2
Reviewer 1 Report
The authors have tried to answer the questions of the reviewer in this revised version.
Nevertheless, the reviewer believes that there serious problems with the experimental setup and/or procedure that lead to non-physical results (variation of the volumetric strain for undrained tests on a saturated material).
Author Response
Reply: The authors sincerely thank the reviewer for this constructive suggestion for improving the manuscript. After reviewing the test process and test data, the author found that the volumetric strain measured during the test is caused by the rubber film embedding effect of the sand sample, which only affects the volumetric strain of the sample during the test. In this paper, the analysis of the volumetric strain of the sample also further describes the variation law of the rubber film embedding effect of the sand sample under cyclic loading. The author has modified the content related to volumetric strain in the article.
According to this comment, on page 11, lines 307-315 have been modified, and the specific content is as follows: “The addition of geogrid can reduce the excess pore water pressure of the sample, but it has a slight increase in the rubber mold embedding effect of the sand sample. As the loading frequency increases Improving the rubber mold embedding effect gradually weakens.”.
Page 9, lines 252-261, “In the text, when the volumetric strain is positive, it means the sample volume expands, when it is negative, it means the sample volume shrinks. This test is an undrained test, and the increase in sample volume strain is caused by the embedding effect of the sand sample membrane. Some scholars have found that when the sand sample is consolidated, the rubber film will be embedded in the pores of the surface of the sample, and the greater the effective stress, the greater the amount of rubber film embedded. During undrained shearing, as the pore water pressure increases, the effective stress decreases, and the embedded amount of the rubber film decreases, that is, part of the water in the sample squeezes out the rubber film, resulting in the sample under undrained conditions. The phenomenon of volumetric strain growth occurs [56].”.
Page 9, lines 265-268, amended to read “It shows that the addition of geogrid can slightly aggravate the rubber mold embedding effect of sand samples, and the rubber mold embedding effect will gradually appear as the number of cyclic load loading increases.”.
Page 11, lines 296-302, amended to read “By including reinforcement, the accumulative volumetric strain increased slightly, and the more the reinforcement layers, the greater the increment. The reinforcing effect coefficients by 5.3% when including three layers of reinforcement, but the value is 2.6% and 1% when including 2 layers and 1 layer of reinforcement respectively. Therefore, the addition of geogrid can slightly aggravate the rubber mold embedding effect of sand samples, and the more layers of geogrid are added, the more obvious this phenomenon will be.”.
Page 13, lines 373-376, amended to read “This shows that the lower the frequency, the more significant the rubber mold embedding effect of the sand sample is. When the frequency is 1Hz and 5Hz, the volumetric strain of the sample under 10,000 cyclic loads is 0.156% and 0.057%, which is a difference of 0.099%.”.
Page 15, lines 412-415, amended to read “Under the same loading frequency, the addition of geogrid can slightly aggravate the rubber mold embedding effect of sand samples. For samples with the same number of reinforcing steel layers, the lower the loading frequency, the more significant the rubber mold embedding effect of the sand samples.”.
Page 17, lines 534-535, amended to read “Cen, C. L., Zang H. M., 2000. Some problems in triaxial test on saturated sands. Chinese Journal of Geotechnical Engineering. 22(06):659-663 (In Chinese).”

Reviewer 3 Report
Dear Authors, thank you for the updates. Just a minor remark to your conclusions 'add one more paragraph before the conclusion points where you indicate shortly what the study was about and what are the conclusions'. Just keep an eye that abstract and conclusions have similar outcome of your research.
Author Response
Reply: The authors sincerely thank the reviewer for this constructive suggestion for improving the manuscript. This has been revised in the conclusions.

Reviewer 4 Report
The paper is ready to be accepted
Author Response
Reply: The authors sincerely thank the reviewer again for helping us to improve the manuscript.

Round 3
Reviewer 1 Report
The authors have provided to some extend a satisfying response to the reviewer's questions and concerns.
The few sentences that have been added should be carefully checked for grammatical and spelling validation.
Author Response
The authors sincerely thank the reviewer for this constructive suggestion for improving the manuscript. After reviewing the test process and test data, the author found the embedding effect of the rubber membrane of the sand sample causes the volumetric strain of the sample to change during the test, the analysis of its volumetric strain can further use to describes the embedding effect under cyclic loading condition.
The author has modified the content related to volumetric strain in the article, as follows:
According to the comment, the contests on page 10, lines 288-290 have been modified. “The addition of geogrid can reduce the excess pore water pressure of the sample and increase slightly the embedding effect of the sand sample. Furthermore, the embedding effect gradually weakens as the loading frequency increases.”
On page 9, lines 245-254, “During the test, When the volumetric strain is positive, it means the sample volume expands, while the sample volume shrinks when it is negative. During the undrained test, and the increase in volume strain is caused by the embedding effect of the sand sample membrane. Some scholars have found that when the sand sample is consolidated, the rubber membrane will be embedded in the pores of the surface of the sample, and the greater the effective stress, the greater the amount of rubber membrane embedded. During undrained shearing, as the pore water pressure increases, the effective stress decreases and the embedded amount of the rubber membrane decreases, that is, part of the water in the sample squeezes out the rubber membrane, resulting in the sample under undrained conditions. The phenomenon of volumetric strain growth occurs [56].”
On page 9, lines 258-260, “It shows that the addition of geogrid can slightly aggravate the embedding effect of sand samples and it will gradually appear as the number of cyclic load loading increases.”.
On page 13, lines 361-364, “This shows that the lower the frequency, the more significant the rubber membrane embedding effect of the sand sample is. When the frequency is 1Hz and 5Hz, the volumetric strain of the sample under 10,000 cyclic loads is 0.156% and 0.057%, which is a difference of 0.099%.”.
On page 14, lines 399-402, “Under the same loading frequency, the addition of geogrid can slightly aggravate the rubber mold embedding effect of sand samples. For samples with the same number of reinforcement layers, the lower the loading frequency, the more significant the rubber mold embedding effect of the sand samples.”
